# Towards a Trusted and Unified Consortium-Blockchain-Based Data Sharing Infrastructure for Open Learning—TolFob Architecture and Implementation

**Jun Xiao** [1,*], **Yi Jiao** [2], **Yin Li** [2] and **Zhujun Jiang** [3]

1 Shanghai Engineering Research Center of Open Distance Education, Shanghai Open University, Shanghai 200433, China
2 Blockchain Technology Institute, Shanghai Academy of Science and Technology, Shanghai 201112, China; Jiaoyi@fudan.edu.cn (Y.J.); liyin@sscenter.sh.cn (Y.L.)
3 Department of Educational Technology, College of Education, Shanghai Normal University, Shanghai 200234, China; zhujunjiang_5@163.com
* Correspondence: ecnuxj2003@163.com

**Abstract:** Open learning is now facing a complex higher education ecosystem that involves a variety of heterogeneous information systems and comprises decentralized stakeholders, such as universities, professors, students, and software vendors. Authentic, non-repudiable, and fast available data sharing among open learning information systems and stakeholders is a key issue that remains unresolved. To solve this problem, this paper proposes a consortium blockchain extended architecture featuring integration and cross-chain functions to provide a unified and trusted data-sharing infrastructure for open learning. The overall architecture consists of three elements: a blockchain-integrated open learning scenario schema; a blockchain-integrated open learning application model; and a pragmatic blockchain integration framework. The proposed blockchain integration framework is implemented based on Hyperledger Fabric 1.4. A trusted open-learning behavior and achievement management application is developed as a proof-of-concept which integrates two educational institutions' four productional learning systems into a blockchain network and has stably run over six months. A suite of experiments is designed and executed to verify our blockchain system's viability and scalability. The test result shows the implementation of the blockchain system is competent for the production environment and outperforms related works investigated. However, it does have limitations and optimization potential, which will be studied in the future.

**Keywords:** distance education and online learning; blockchain; architectures for educational technology system; post-secondary education

## 1. Introduction

Today, open learning is facing new problems as the evolution of higher education has come up as an increasingly complex ecosystem, which comprises various stakeholders such as universities, faculties, students, higher educational institutions, governmental sectors, employer entities, and EdTech software companies [1,2]. In open learning cases, most of the stakeholder organizations have established their proprietary informational systems [2,3], such as the very popular Massive Open Online Courses (MOOCs) platforms of Udacity [1,3,4], Coursera [5], and edX [6], as well as various on-line or remote learning platforms in large universities (e.g., MIT, Stanford, etc.) or open universities [1,2]. As a result, professors giving lectures utilizing diverse remote teaching tools and students carrying out learning processes on different HEI software platforms are now very common states. These proprietary HEI informational systems are usually heterogenous, decentralized, and difficult to interact with. Furthermore, numerous open learning resources (and open educational resources in a broader sense) are dispersed and separated in these

isolated, globally distributed HEI information systems. It is difficult to achieve effective sharing of open learning resources and, hence, a great waste of educational resources can occur [2,7–10].

An effective interaction mechanism plays a paramount role in the open learning sector as well as the entire education ecosystem [11]. However, interoperability issues, or, more concretely, the question of authentic, non-repudiable, and quickly available data sharing among related open learning information systems and stakeholders is a key issue that remains unresolved.

The paramountcy of the interaction mechanism in an open learning ecosystem is embodied in many other scenarios too. For instance, in joint-cultivation projects, universities need to exchange and record students' credit and related exam details. When hiring personnel, employer companies need to carefully check and confirm students' credentials to avoid fraud and falsification. It would be of value to observe applicant student's detailed information such as behavior records and learning process achievements, if these data can be provided and ensured to be authentic, and to determine whether the candidate is qualified with claimed competencies and character [1,12,13].

In these scenarios, students' important data such as credentials (transcripts, certificates, e.g., diploma, degree, training, internship, etc.), learning process behavior records, and study achievements records need to be carefully produced, issued, preserved, and shared among different stakeholders [14,15]. Moreover, the sharing process needs to be easy, fast, and available on-demand. Meanwhile, security, authenticity, and confidentiality should be guaranteed. Such a data sharing mechanism has been and still remains a key issue not yet properly resolved [16].

Numerous technologies have been devised to mitigate this data integration problem, e.g., data interface, web services, role-based access control, etc. However, these classic or traditional technologies usually aim at merely one or several aspects of the complex data exchange issue. It heavily depends on system designer's expertise and proficiency to determine which technology or technologies should be adopted or combined during system implementation. This may cause customized, special-purpose, and cumbersome solutions rather than comprehensive and general-purpose ones. It would deteriorate in a team coding situation. The design process would be complicated, time-consuming, and error-prone. Consequently, software systems are easy to falsify through back-end databases, and they are hard to detect and prevent.

In the past decade, blockchain has become a promising approach to solve this problem [17]. It features a novel computing paradigm that combines four categories of technology: distributed storage; peer-to-peer network communication; distributed consensus mechanism; and cryptography [18,19]. Originally, it comprised the implementation of the cryptocurrency Bitcoin. It has quickly evolved as a technical platform, a systematical and comprehensive data exchange solution for heterogeneous systems. Many blockchain implementations have been developed, to enumerate, such as Ethereum, R3 Corda, Hyperledger Fabric, Libra, etc. [19]. It has attracted numerous researchers' attention and has been introduced to many fields, such as integrity verification, anti-piracy, supply chain management, and medical and health, not to mention education [20]. Blockchain can be implemented in education and can enhance higher education, and developing an "educational infrastructure" to support online learning is part of this [1].

Figure 1 concisely summarizes the key stakeholders and challenges of data sharing in our open learning context, key technologies of blockchain, and typical open learning applications that could be enhanced as "trusted" with blockchain technology. Obviously, there are difficulties and challenges in adopting blockchain to solve the open learning trusted data sharing issue.

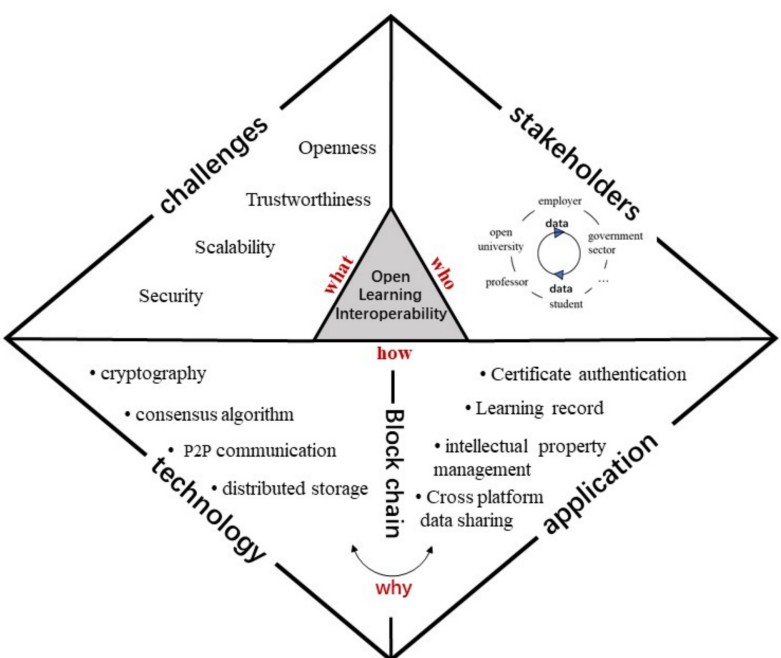

**Figure 1.** Depiction of data sharing challenges and blockchain enhanced applications in open learning.

This study is conducted to respond to the following research question: What would a unified, trusted and blockchain-based data-sharing infrastructure need to be to solve the interoperability issue in an open learning ecosystem?

We further decompose the research question into three sub-questions:

1.  What would the open learning scenario be for the introduction of such a blockchain infrastructure? (scenario schema);
2.  What would the open learning application be for the introduction of such a blockchain infrastructure? (application model);
3.  What would the integration means be to provide an infrastructure for HEI information system integration into blockchain? (integration framework).

Based on a detailed literature review and decades of software development expertise in EdTech, our methodology is to propose a consortium blockchain-based architecture which puts a business process schema, a conceptual model, and a pragmatic developing framework in a synergic usage as a potential solution to the research question. We try to respond to the research question from a view combining both software architecture (supporting system integration) and open learning business features. As presented in this work, we provide an implementation that leverages consortium blockchain technology and makes further extensions for proprietary HEI systems integration, meanwhile achieving trusted open learning data-sharing management and integrity verification.

The main contribution of this paper is as follows:

(1)  Propose a consortium blockchain-based architecture serving as a trusted and unified data sharing solution among decentralized open learning ecosystems;
(2)  (a) Design and develop a pragmatic extended blockchain network, using Hyperledger fabric 1.4.4 LTS, and using cache database to optimize blockchain system data processing performance; (b) design and develop a "trusted open learning behavior and achievement management" application as a proof-of-concept of the proposed architecture;
(3)  Run the proof-of-concept application for 6 months with data from production systems and conduct a series of tests on its performance and scalability to analyze and verify the extended blockchain network.

It is worth mentioning that our proposed blockchain infrastructure is suitable not only for the open learning sector but also the higher education domain in general, provided that the application scenario is based on the mature adoption of information systems. An important consideration of selecting the open learning sector as our research object is that this sector bears a more vivid informational feature compared with other ones.

The remainder of this paper is organized as follows: Section 2 elaborates on blockchain technology and related work in education field; Section 3 explains the research methodology with the proposed architecture; Section 4 clarifies the results obtained in architecture implementation and proof-of-concept; in Section 5, discussions are covered by experiments on performance, scalability, and comparison with previous work; finally, Section 6 concludes this paper with future work.

## 2. Blockchain Technology and Related Work

### 2.1. Blockchain Technology

This section will elaborate on blockchain technology. As blockchain has come up as a complicated technology, due to space limitation, we will not make a thorough technological examination herein. Rather, a concise depiction necessary for this work will be presented.

Bitcoin is the origination and first application (maybe the most successful one) of blockchain technology. In the years 2008–2009, Satoshi Nakamoto (pseudonym) improvised and released this cryptocurrency with the aim of a decentralized payment system, which can run anonymously and autonomously without intermediation. To fulfill this objective, four categories of computing technologies were creatively integrated in one application for the first time: distributed storage; cryptography; peer-to-peer (P2P) communication; and consensus algorithm [19,21]. Its basic data structure is called block, which consists of a header part and a content part. Blocks are identified using a hash function and can be linked chronologically as an ordered list, called a "chain" structure. End-user nodes issue transactions using an asymmetric cryptography scheme (a pair of public key and private key, which ensures non-repudiation). Transaction data are packaged into blocks and made hashes (un-tampered). Blocks are propagated to all participant nodes, then stored and listed to chains locally. List sequence is maintained in collaboration by all participant nodes through a "consensus reaching" scheme, which ensures the prevention of malicious modification of previous blocks [22]. For more in-depth technical details of Bitcoin, interested readers may refer to [21].

Following the boom of Bitcoin, a diverse range of blockchain implementations were developed. Many are still cryptocurrency applications without computing capacity and sometimes are called Blockchain 1.0 by some researchers. Some other blockchain variants are designed with computing capacity and are correspondingly called Blockchain 2.0 [17]. Ethereum and Hyperledger Fabric are the two most notable representatives to date. Ethereum introduces a Turing-complete programming language and support smart contract coding capacity with Solitary. It releases the token of "Ether" and supports distributed application (DApp) development. Hyperledger Fabric defines "chain-code" as a smart contract and supports Go and Java programing. Fabric is module-designed, extensible, and suitable for non-financial industry applications [18,22].

In early category blockchains, all nodes can freely participate in the "consensus reaching" process, called "permissionless" mode, or public chains, such as Bitcoin and Ethereum. Public chains have an obvious weakness of too much overhead. Later, new implementations made constraints on nodes' participation of consensus reaching, called "permissioned" mode, or consortium (federated) chain, such as Fabric. Generally, these consensus nodes represent different organizations. If all the consensus nodes belong to one organization, it forms a weak version called private chain [23].

Currently, blockchain is still an active field with new technology features continuously being put forward. Moreover, new problems arise, e.g., performance issues and interoperability issues. Blockchain's performance efficiency has received much criticism. Opponents argue that both block and list are preliminary store structures with low efficiency compared

to database technologies [2,4]. Further, with fast adoption and heterogeneous implementations, interoperability or standardization issue deteriorates [18].

Our goal is to utilize blockchain technology to alleviate the cumbersome data sharing issue in the open learning field. We give our observations and evaluations as follows:

(1)	Public chain is more suitable for internet applications, and is not organization-friendly;
(2)	For an enterprise lever application, consortium chain performs better and is more suitable for production system development.

### 2.2. Related Work

The literature review shows that a variety of blockchain research and applications have been conducted in the education domain. In this section, we summarize some research works that are related to our work.

In [1,3,17,22,24], authors regarded blockchain as one of the state-of-the-art disruptive technologies, which could change society and create a new social and financial ecosystem. Authors exploited the capacities of blockchain in education and discussed various application scenarios such as certificate digitization and counterfeiting-proof, enhancing and motivating lifelong learning (lifelong learning record verification), sharing documents between institutes, identity verification, digital right management, protection of intellectual property, automating administrative tasks, promoting job matching, etc.

In [11], authors put forward an Ethereum-based model of confidence for higher education management. The model utilizes Ethereum as an academic cryptocurrency and a tool to manage transactions of content, teaching and competencies, which are assessed with consensus by stakeholders to eliminate the gap between the academic and the working world. Authors developed an Ethereum-based prototype and conducted experiments to validate the proposed model. In [20], authors proposed a blockchain platform for lifelong learning records management. It utilizes Ethereum with a smart contract that connects the learning logs of students across different institutions into a single, public, and immutable ledger.

The paper [25] presented a blockchain-based architecture for a ubiquitous learning environment. This work suggests that blockchain provides data exchange within the decentralized topology. However, this paper does not cover implementation details and experiments. In [26], authors proposed a permanent distributed record associated with reputational rewards in OpenLearn project. They implemented an open blockchain platform using Ethereum (permissionless, private) and made initial trials. However, the work did not propose any architecture nor test-based results.

The paper [27] discussed a blockchain-enabled School Information Hub (SIH) with conceptual framework, initial design, implementation on Hyperledger Fabric, and a case study using Kenya's school system. The work primarily focuses on testing the effectiveness of the proposed Fabric-based SIH system and does not carry any test-based result nor experimental analysis. It also does not mention cross-chain issue and the architecture scalability as well.

Some published works that are closely related to our work are [2,4,16,28]. Authors adopted a consortium blockchain and implemented a prototype with Hyperledger Fabric 1.4, which was similar to our work. However, the proposed scheme involved neither HEI information systems integration framework nor cross chain consideration. The prototype mainly presented some smart contracts in algorithm and the performance is discussed in computational cost instead of system running data.

The case is similar in [16]. The architecture proposed in [16] aims to provide a secured sharing solution for students' credentials. It comprises five stakeholders, an Ethereum blockchain implementation with nine smart contracts, and a distributed off-chain file storage. Authors conducted numerous tests to prove its performance and viability. However, the main focus of this architecture was the roles, functionalities, and business processes of different stakeholders. It did not involve data integration or sharing from HEI existing legacy information systems and therefore is a simplified scenario compared to our work.

The paper [28] proposed a blockchain-based trusted data management scheme called BlockTDM. The authors aimed to provide a general, flexible, and configurable blockchain-based paradigm for trusted data management in edge computing environments, but not in the education sector. Moreover, the implementation was a specific application of a blockchain-based data management system, rather than a data integration and sharing framework, as in our work.

Our work distinguishes from the existing related works as follows. First, besides the open learning business model view, we also hold a software architecture perspective. We focus on a more pragmatic scenario of data integration among multiple existing legacy systems, as is the case in most of the current open learning ecosystems. Therefore, consortium blockchain serves as a better candidate. This enables our architecture to have a capacity for a wider scope of data category integration. In fact, almost all data stored in legacy HEI information systems can be entered into blockchain and shared by permissioned organizations, not only transcripts, credentials, but also learning achievements, learning process records, etc. Second, our architecture takes the chain interoperability issue into consideration and proposes theoretical protocols and initial implementation. This cross-chain-containing solution makes our architecture more scalable and general-purpose. Third, to validate the proposed architecture, we have developed and deployed an extended Hyperledger Fabric production system, which integrates four legacy HEI open learning systems and has stably run for 6 months. We conducted numerous tests and experiments during this stage.

## 3. Methodology: Proposed Architecture

A business schema of a blockchain-integrated open learning scenario, an application model of blockchain-integrated open learning, and a framework of blockchain integration are used as methodological approaches in this research. These three can respectively function as scenario architecture, application architecture, and technology architecture for open learning data sharing, and jointly constitute an overall architecture that serves as a trusted and unified data sharing solution for open learning ecosystem, as named "Trusted open learning process and achievement management framework on blockchain" (TolFob) in our work. The overall architecture comprises two different abstraction levels: Level 1 involves conceptual depictions for theoretical research, which includes the scenario schema of open learning and its business process abstraction with blockchain integration (Section 3.1), as well as the conceptual application model of open learning with blockchain integration; while Level 2 is the guideline and means for pragmatic software development, i.e., the framework for trusted consortium blockchain-based open learning system integration and implementation (Section 3.3).

### 3.1. Business Schema

In order to provide a trusted, unified, blockchain-based data-sharing infrastructure for open learning, our first step concerns attempting to depict the business schema of an open learning scenario with the integration of blockchain. That is, we need to discern, with the introduction of such a blockchain infrastructure, what the open learning scenario will be. This section presents the proposed business schema for consortium blockchain enhanced open learning scenario and abstraction of its business process. Figure 2 illustrates a conceptual data-sharing snapshot of the open learning ecosystem studied in our work. The initial solution in our proposal is based on only one single consortium blockchain network. However, considering the increasing adoption of blockchain technology, it would be the case that different groups of HEIs and government sectors construct different blockchain networks with various implementations. Therefore, our enhanced proposal is strengthened by an extensional component of cross-chain mechanism to handle the forthcoming blockchain heterogeneity, which will be discussed in the next section.

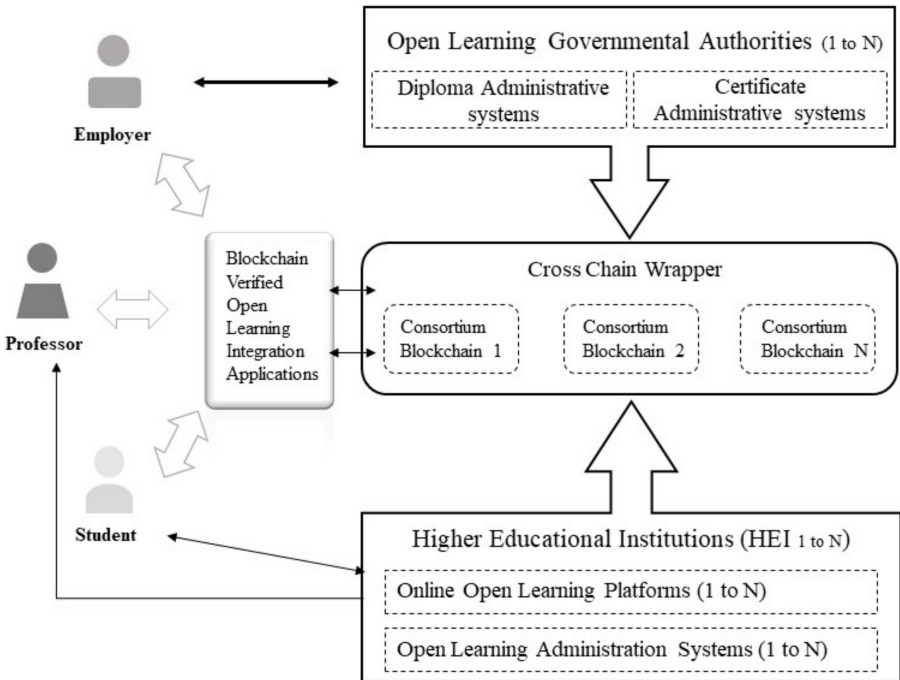

**Figure 2.** A snapshot of open learning ecosystem data sharing scenario.

In general, the proposed open learning scenario business schema could be formalized as a 9-tuple: <Is, Sr, Br, Oe | Tn, Bc, Sy, Oc, An>, in which Is denotes proprietary HEI open learning Information system, Sr denotes open learning Stakeholder, Br denotes Behavior, Oe denotes Outcome, Tn denotes Transaction, Bc denotes open learning Blockchain system, Sy denotes Strategy, Oc denotes Off-chain storage, and An denotes blockchain-enabled integration application of open learning.

More concretely, we make further definitions and denotations as follows:

Sr: <Srs, Srp, Sre, Sro, Srg>, denotes student, professor, employer, HEI open learning institution (open university), and government sector respectively. To propose the architecture, we first discern and define these five fundamental categories of stakeholders;

Is: <Isa, Isp, Isn, Isd, Isc>, denotes educational administrative system, diploma-oriented online teaching system, non-diploma-oriented online open course system, government's diploma administrative system, and certificate administrative system as well. Clearly, in an open learning ecosystem, data sharing is heavily dependent on information system interaction. Therefore, we further discern and define five categories of HEI proprietary information systems;

Br: <Brs, Brp, Bre>, denotes Behaviors performed by student, professor, and employer respectively. It is somewhat difficult to accurately define and categorize all the practical open learning behaviors; herein, we name some typical examples: for students, separate and statistical data of <timestamp, time length, content> of online learning activities, such as course-taking, video watching, discussion and quizzes; for professors, issuing data of <timestamp, content> of scores, etc.;

Oe: <Oen, Oep>, denotes Outcome non-available, and Outcome presented by open learning, e.g., student's essay, picture, model (in the digital version), etc.;

Tn: <Tnt, Tnb>, denotes Transaction intrigued by HEI proprietary learning system and blockchain respectively; e.g., Tnt: log-in, log-out, upload, download, post, submit, etc.; Tnb: hash, public key encryption, etc.;

Bc: <Bcb, Bcc>, denotes specific Blockchain implementation and cross-chain middleware respectively;

Sy: <Syc, Syo, Syp, Sys, Syh> denotes strategy with or without cross chain, with or without off-chain storage, with or without privacy, with or without security, with or without hybrid tradeoff (e.g., privacy-aware, and security enhancement) respectively;

Oc: denotes Off-chain storage;

An: <Ani, Anc, Ano>, denotes blockchain-enabled new applications, e.g., integration verification (data provenance and counterfeit), credential data sharing (diploma, certificate, and behavior record), outcome intellectual property (IP) management, etc.;

With these defined notation sets, the proposed open learning scenario schema could be described as a suite of open learning business process sets, as illustrated in Figure 3.

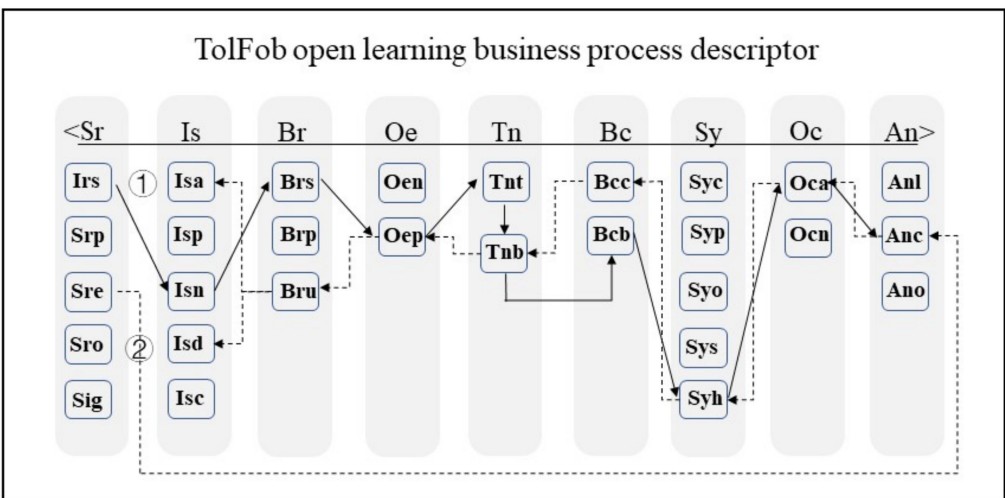

**Figure 3.** TolFob architecture: business process descriptor and two examples.

Without loss of generality, a normal business process description of the proposed schema could be: An open learning stakeholder (Sr) performs an open learning behavior (Br) on a proprietary HEI open learning information system (Is), and produces an open learning outcome (Oe), and intrigues a transaction (Tn) which is automatically submitted to the blockchain (Bc) with a selected strategy (Sy), and reflects to a blockchain-enabled new open learning application (An).

To clarify, here we select two typical open learning business processes and specify them in detail.

Open learning Business process 1: (line ① in Figure 3.) A trusted, secured open learning behavior and outcome sharing process without privacy protection.

A student stakeholder (Srs) logs in a non-diploma-oriented online open course system (Isn), studies an on-line construction modeling course (e.g., Sketch-up) (Brs), submits (Tnt) a series of Sku models (Oep) later, with the hash of blockchain (Tnb), the learning record and Sku models being entered into a blockchain system (Bcb) with the hybrid strategy (Syh) of "without cross-chain, with off chain-storage, without privacy, with security" (Oca), and finally ended with the integration verification application (Ani) and outcome IP management application (Ano).

Open learning Business process 2: (line ② in Figure 3.) A trusted, secured certificate sharing process with privacy protection.

An employer stakeholder (Sre) logs in a credential data sharing application (Anc) to check a candidate's diploma or certificate. The diploma was stored in an off-chain storage (Oca) with a hybrid strategy (Syh) of "with cross-chain, with off chain storage, with privacy, with Security" (Oca). The hash of diploma (Tnb) was stored in a blockchain system with cross chain functionality (Bcc), by an open university staff (Bru) on an educational administrative system (Isa) and the government's diploma administrative system (Isd).

### 3.2. Application Model

When the business schema of a blockchain-integrated open learning scenario is abstracted, our next step is trying to depict the application model. That is, we need to discern, with the introduction of a blockchain infrastructure, what the open learning application will be. This section presents our proposed conceptual model for a consortium blockchain-integrated open learning application. The conceptual model for integrated and verified open learning applications is shown in Figure 4. The proposed model comprises eight tiers that can be further divided into three layers: a consortium blockchain layer; a cross-chain layer; and a trusted open learning application layer. The eight tiers are enumerated as follows from a top-down point of view: trusted open learning application tier; HEI learning system business abstraction tier; cross-chain tier; adaptation tier; contract tier; security tier; storage tier; and network tier.

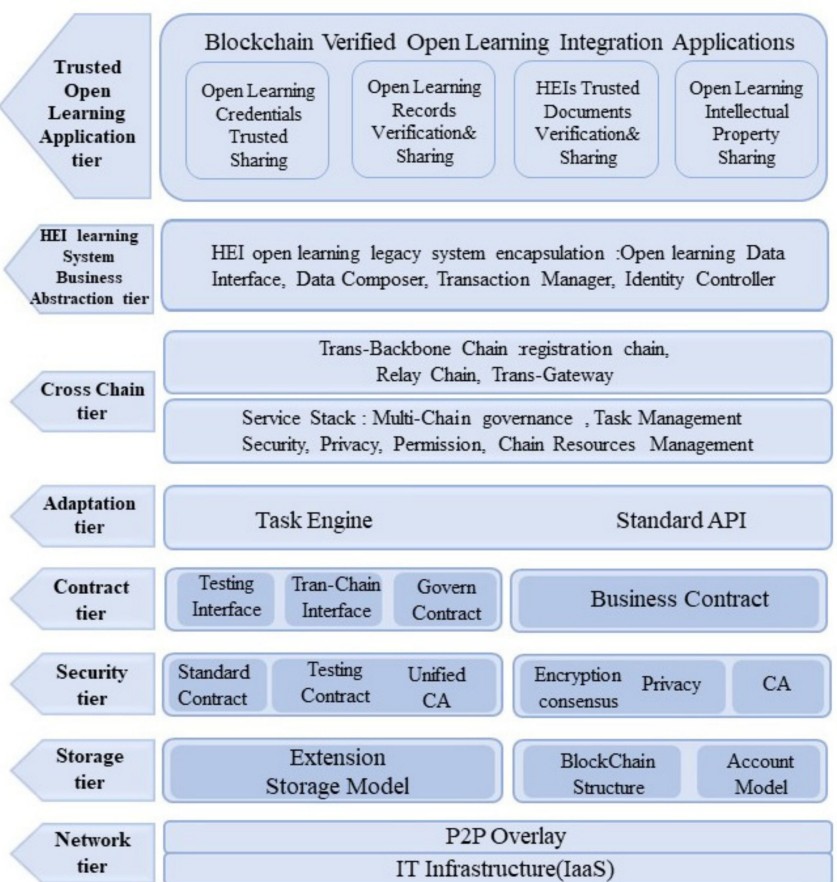

**Figure 4.** The conceptual 8-tier model for consortium blockchain integration application.

The core components of the proposed model are as follows:

(1) Trusted open learning Application layer: This layer consists of the entire contents of the HEI learning system business abstraction tier and trusted open learning application tier.

Trusted open learning application tier: refers to a suite of open learning applications based on authentic data exchange that collects, stores, and manages designated data from existing HEI learning systems, such as credential sharing applications, learning achievements sharing applications, etc. These applications are defined as trusted and permissioned open to stakeholders.

HEI learning system business abstraction tier: this tier comprises four components: legacy HEI learning system data interface; open learning data composer; transaction man-

ager; identity controller. Legacy HEI learning system data interface is executed on the legacy HEI system side, which encapsulates designated data and sends to data composer component in accordance with agreed formats. The open learning Data composer component constructs metadata model and meta mapping, receives open learning data from legacy HEI learning systems, and makes the policy of correlation, combination, standardization, etc. The transaction manager component works on both application and legacy HEI learning system sides, which detects and analyzes transactions invoked and decomposes to legacy HEI learning system interface as queries. Identity controller components work as unified ID for data correlation among legacy HEI learning systems, applications, and blockchain storage;

(2)　Consortium blockchain layer: This layer consists of the entire contents of the network tire as well as part contents of the storage tire, security tier, and contract tier.

Network tier: refers to normal IT infrastructures including the common service of IaaS that provides P2P overlay networks for a consortium blockchain.

Storage tier: includes a blockchain structure and an account model that can joint construct a distributed ledger for the network.

Security tier: comprising an encryption component, a consensus component, a privacy component, and a CA component to provide features of tamper-proof, immutability and non-repudiation of data operation.

Contract tier: provides the capacity of customizing business contract that supports the blockchain-entry function of selected business data.

This generic layer depicts the fundamental components of formulating a consortium blockchain network;

(3)　Cross-chain layer: This layer consists of the entire contents of cross-chain tier and adaptation tier, as well as part of the storage tier, security tier, and contract tier.

Cross-chain tier: This tier is defined as comprising a service stack and a trans-backbone chain. The service stack provides necessary service for heterogenous consortium blockchain interaction, mainly including: multi-chain governance service, task management service, security management service, privacy management service, permission management service, chain resources service. The trans-backbone chain is defined as three core components: registration chain; relay chain; trans-gateway. The service stack and Trans-backbone chain together construct a unified facility for chain-chain interoperability.

Adaptation tier: This tier defines two core components of the task engine and standard API, which works on the consortium blockchains side to provide necessary adaptation to the cross-chain tier.

Contract tier: This tier provides three core components: standard contract; testing contract; governance contract.

Security tier: This tier comprises three core components: Testing interface; Trans-chain interface; unified CA.

Storage tier: the core component of this tier is extension storage model, which means an extra data storage mechanism acting as either off-chain or in-line chain as cache, to improve performance efficiency of the blockchain.

This conceptual eight-tier model outlines the generic components and logical relations for the complex blockchain-based open learning sharing application design. To date, the model carefully takes blockchain interoperability into consideration, and proposes a set of protocols to construct a unified and trusted infrastructure.

*3.3. Integration Framework*

After the depiction of scenario schema and application model, our final step is trying to design a pragmatic software development framework as a guideline and unified means for implementation. This section presents our proposed blockchain integration framework for trusted open learning application development. The aim of the framework is that open learning system designers can have a unified and comprehensive means to collaborate,

without too much concern over issues such as openness, trustworthiness, or scalability. These features are ensured by the nature of the blockchain network.

Figure 5 shows a concise view of the proposed framework that consists of three divisions: the trusted Open Learning Integration Application (OLA); the Pragmatic Blockchain System for open learning (PBS); and the Legacy HEI open learning System (LS). PBS stands at the center of these three divisions. It consists of six core components: Open learning Data Collecting, Chain-Entry Data Standard Preprocess, Unified Chain-Entry Interface, Chain-Entry/Enquiry/Cross Chain Middleware, Off-Chain Storage, and the back-end Consortium Blockchains.

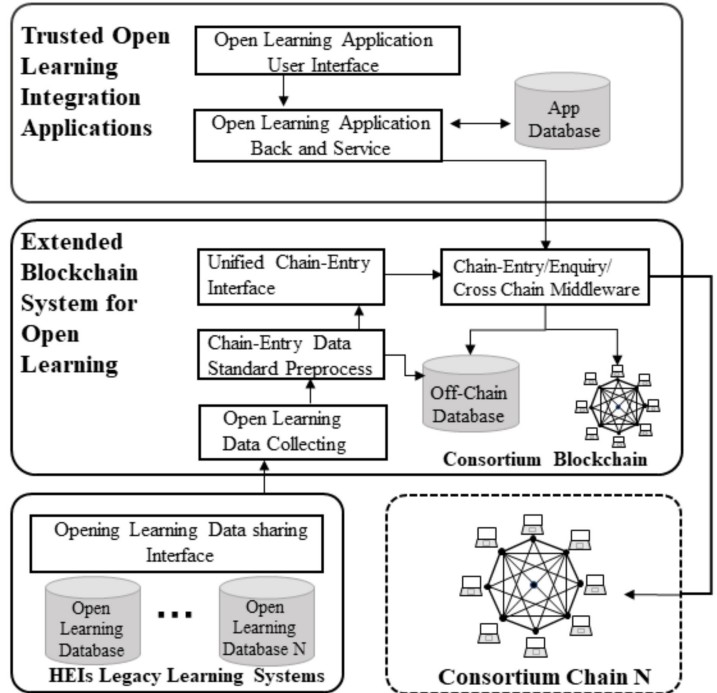

**Figure 5.** Framework for pragmatic system development.

A general process works in the following three steps. Step 1: the OPA designer analyzes the overall scheme of data-to-be-shared in trusted open learning application and forms open learning data specification; Step 2: the LS designer implements Data Sharing Interfaces according to the trusted open learning data specification; Step 3: the PBS designer implements legacy HEI open learning system's open learning Data Collecting and a series of Chain-Entry transactions. Step 2 is comparatively trivial. We will come to discuss Step 1 and 3 in the following.

In Step 1, the core work is to define an open learning data scheme and open learning Meta-data for a new trusted open learning application. The fundamental work before the Chain-Entry Data Standard Preprocess is defining of a unified open learning data scheme that governs all the data from participating legacy HEI open learning system, thus achieving an unified open learning data process and integration.

The open learning data scheme comprises three levels of considerations:

(1) Open learning Data format level: open learning business data format is defined using unified JSON specification;
(2) Open learning Data semantic level: define the same naming for different fields with the same meaning; for different business types, define fundamental data structure and extension data structure; adopt unified open learning Meta-data management to facilitate semantic recognize by code;
(3) Open learning Data security level: for sensitive open learning data, define data masking and encryption algorithm standard, including specifications related to data

security and authorization, as well as digital signature and verification specification for Chain-Entry data.

Open learning Meta-data defining for new trusted open learning application is illustrated as follows:

A suite of open learning Meta-data is defined in this proposed pragmatic framework. Some core ones include legacy HEI open learning platform record, teacher record, student record, learning behavior record, learning outcome record, learning activities related to records, etc. A legacy HEI open learning platform record consists of university ID, platform ID, student records, etc. Student records and teacher records mainly consist of profiles. A behavior record consists of system log attendance, start learning or close learning a course, learning time span, etc. A legacy HEI open learning outcome record consists of uploaded learning achievements, grades and evaluations, etc. A legacy HEI open learning activity related to a student record can include: registration of student in learning platform; double-checking a student's identity with biometric data such as face image; enrollment in a class, etc. Similarly, activities related to teacher record, behavior, and outcome include teacher registration, selection of a course, upload or update submission, etc.

In Step 3, the core work is to define cross-chain Interface and Chain-Entry Interface.

Cross-chain Interface specification is illustrated as follows:

Open learning process Smart contract, block, and open learning user account are abstracted as blockchain resources in a unified method. Access to blockchain resources can be reached at any site of a cross chain system through the combination of network, chain ID, and resource name, which forms the access address of the unified resources. The addressing path can be defined as: [Network]/[specific chain ID]/[Resource Name].

The Atomic cross-chain access implementation would be: leveraging HTTP Restful interface visit cross-chain path, and supporting resource access in cross-system with HTTP URL.

Chain-Entry interface is illustrated as follows:

Chain-Entry interface adopts a RESTful paradigm.

A simplified open learning Chain-Entry process specification would be:

(1) Define unified open learning Chain-Entry data schemes, including time-stamp, CA signature, business key-value, and business data value;

(2) In accordance with the defined open learning Chain-Entry data scheme, define a suite of standard open learning smart contracts for data Entry-Chain and Chain-Enquiry, which support different implementations for specific consortium blockchains;

(3) Based on the suite of standard open learning smart contracts, define unified open learning data Entry-Chain and open learning Chain-Enquiry API that shields differentiations of specific blockchain implementations, thus achieving cross-chain interoperability, enabling integration and scale for future consortium blockchains.

During the open learning Chain-Entry process, the off-chain storage issue should be handled to store the bulk of open learning business data.

## 4. Results

This section presents our implementation and proof-of-concept application to serve as results of the proposed architecture.

As stated before, the TolFob architecture is specified to bring trust data sharing for collaborative open learning systems. We consider the following requirements when implementing the architecture: (i): Trustworthiness: open learning provenance data must be collected and stored, immutably, and must be trustworthy; (ii): Transparency of provenance data sharing: Blockchains are fundamentally transparent, where data and interactions are visible to all open learning participants in the blockchain network; (iii): Privacy: open learning provenance data should be shared between authorized personnel; (iv): Interoperability: open learning provenance data collected from legacy HEI open learning systems should be easily integrated.

The implementation of the TolFob integration framework is shown in Figure 6. It is divided into three modules: on-chain module; off-chain module; and new trusted open learning application module. The on-chain module was implemented using Hyperledger Fabric 1.4.4 LTS platform. To store and retrieve the open learning information in the blockchain, it is necessary to use chaincodes implemented using the Go programming language. The RESTful API Service, the Unified Blockchain API, the Fabric client SDK and off-chain data storage compose the off-chain module which will be fully described in Section 4.1. The new trusted open learning Application module will be described in Section 4.2.

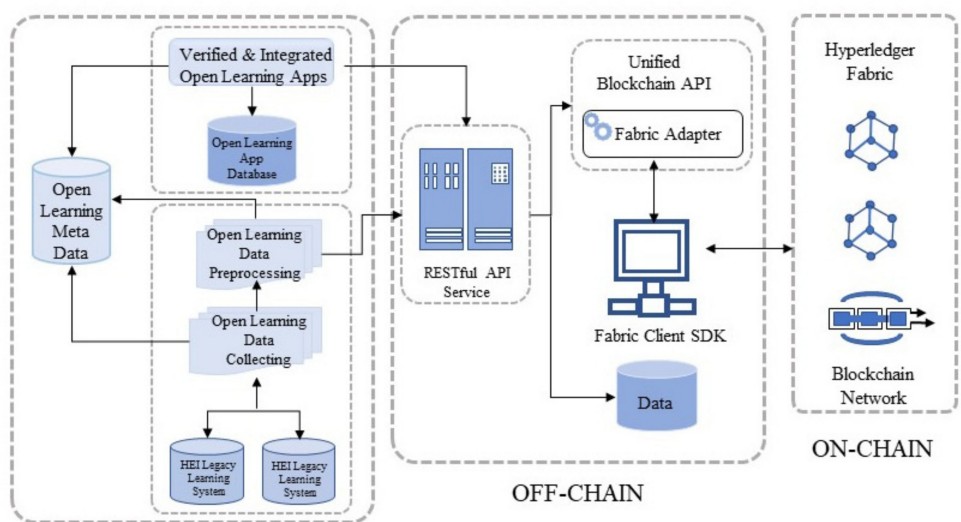

**Figure 6.** Implementation architecture of TolFob.

### 4.1. Implementation of Off-Chain Module

The RESTful API Service in the off-chain module allows TolFob to be integrated with any other HEI open learning platform or application, based on communication via REST web services and HTTP. Its main objective is that open learning platforms and applications can easily store and query open learning provenance data on the blockchain. In the off-chain module, storing and querying provenance data on the blockchain is conducted by Unified Blockchain API, and the specific operation on blockchain can be described by a resource url and params. Table 1 shows an example usage of Unified Blockchain API. The resource *url* in Table 1 contains blockchain network (shouNetwork), blockchain ID (actChain), resource name of smart contract (study) and resource method (insertRecord). The involved open learning Resource specification detail is mentioned in Section 3. The response JSON field is illustrated and shown in Table 2.

The Unified Blockchain API service can configure multiple blockchains, and each blockchain has an adapter implementation. Fabric adapter is implemented using Fabric Client SDK in Java. Both Unified Blockchain API service and RESTful API Service are implemented on Springboot framework. To speed up data access, an off-chain data storage is employed. When HEI learning application submits a record to the blockchain, the RESTful API Service will receive the request and store it in the off-chain data storage for indexing purpose before sending the record to the blockchain. And each record in data storage has a transaction hash field for tracking the corresponding transaction records in blockchain. With help of off-chain data storage, a new trusted open learning application can easily aggregate blockchain records for specific topics, for example, aggregating users' learning activities from the last month. The off-chain data storage is currently implemented on mysql database. Elastic Search and Mongodb will be supported in the near future.

**Table 1.** Request and response of inserting an open learning record on blockchain using Unified Blockchain API.

| | |
|---|---|
| Request | POST http://\{host\}:\{port\}/shouNetwork/actChain/study/insertRecord<br>Header: 'Content-Type: application/json'<br>Body:<br>{<br>    "entityCode": "user",<br>    "entityId":100,<br>    "content":"helloworld"<br>} |
| Response | {<br>"body": {<br>"blockId": 9,<br>"id": 2,<br>"response":<br>"{\"blockId\":9,\"contractMethod\":\"insertRecord\",\"contractName\":<br>\"study\",\"message\":\"\",\"payload\":\"\",\"status\":200,\"timestamp\":<br>1600765829937,\"txId\":\"964d4d3168fe3d344671d297d14fd53da39f7be984554e69<br>acb0f99919830369\"}",<br>"timestamp": "2020-09-22 17:10:29",<br>"txHash":<br>"964d4d3168fe3d344671d297d14fd53da39f7be984554e69acb0f99919830369"<br>},<br>"message": "success",<br>"status": 0<br>} |

**Table 2.** Illustrations of the response JSON field.

| Field | Description |
|---|---|
| Status | status code, with 0 for success and negative for failure Example of text |
| Message | message, with "success" for success and exception message for failure |
| | response contents, in JSON format, with the following field illustrations: |
| | blockID: the height of block |
| Body | txHash: the hash value of transaction entered chain, record in blockchain can be accessed by this hash value; |
| | timestamp: the timestamp for data Chain-Entry; |
| | response: the response message for Chain-Entry execution |

*4.2. Implementation of Application Module*

Before collecting data from legacy HEI open learning systems, a suite of open learning metadata should be defined first. Details about the open learning metadata are described in Section 3. The storage of open learning metadata is built upon mysql database and the metadata managing and maintaining are implemented in an open learning application. When open learning metadata is defined, the data collecting service will read the data source definitions from metadata, where the legacy HEI learning system data accessing interface is defined. Both pull and push mechanisms are supported in the data collecting service and the communication data is in JSON format. The pull mechanism is implemented by a periodically scheduled task to pull data from legacy HEI learning system through a http restful *url*, that is, the legacy HEI learning system must provide the pulling data restful API, and oauth2.0 client credentials protocol is used to protect data resources. The push mechanism is implemented by a restful API running in open learning data collecting service and accepting data record from legacy HEI open learning systems.

After the data record is collected, the HEI open learning data preprocessing service will parse the record and extract the data source and data type field, and read the meta data schema definition from metadata store. The metadata schema defines the rules which

contain field name mappings, indexing fields, filtering fields, and encryption fields and algorithms, and these rules are applied to convert the record in order. After converting the record, the results contain the result data record and indexing fieldsets which will be stored in off-chain storage to speed up data access. According to the data source and data type, the data preprocessing service can read the destination blockchain resource mapping and submit the result record together with indexing filedsets to the RESTful API service and finish ON-CHAIN processing. After the data from diverse legacy open learning systems are put on chain, open learning data sharing and interactions can be easily achieved to all participant open learning applications in the blockchain network.

*4.3. Implementation of a Proof-of-Concept Application*

We designed and implemented a proof-of-concept application named "trusted open learning behavior and achievement management", to validate the proposed architecture and framework. The proof-of-concept application involved three organizations in China, including an open university, a remote-teaching college, and a company that enrolls students from the university and college. We selected and used four legacy HEI information systems, which included a University Diplomat online learning platform (UDP for short), a University Non-Diplomat open course platform (UNDP for short), a university credential management system (UCM for short), and a college credential management system (CCM for short). The former three legacy HEI information systems were owned by the university, and the latter one was owned by the college. These four systems were formerly designed, implemented, and maintained by four different EdTech software vendors. These systems were still normally running production systems. Some of their historical running data are listed below:

Data volume of UDP (as of 30 May 2021): 3154 courses opened, 98,181 teachers and students, 44,000 course resources, a data capacity beyond 7 Terabytes, 376 million formative assessment assignments and self-tests, 32,600 units of student online graduation guidance, 23,968 uploaded papers, 180,000 units of teacher online guidance, 9653 iterations of teaching activity, 750,000 student posts, 1,700,000 teacher posts.

Data volume of UNDP: 19,333 users, 38 courses, 1818 students this year (2021), 4743 h study length, total 40,708 times of learning activity, average 12 times of learning activity per student (activated).

Adopting the proposed architecture and framework, we collaborated with the four EdTech software vendors and jointly designed and developed the new proof-of-concept application. Figure 7 shows a screenshot of the newly developed trusted application.

The reason we selected this scenario roots from a typical open learning data sharing requirement: some registered students (not all) of the college also take part in the university's online open courses and even hope to get the university's diploma as a plus. Therefore, they study and receive credentials from both the college and the university through HEI information systems. The company wants to recruit students from both the university and the college. Therefore, a trusted learning behavior and achievement management application will greatly benefit all the stakeholders of this specific open learning sharing scenario.

Based on the proposed blockchain implementation, we first defined the learning metadata for the four legacy HEI learning systems. The main data scope includes: teacher's profile data; student's profile data; student's learning behavior data; student's learning achievement data; student's courses score; and student's credentials, etc. Some key fields of the open learning metadata are exemplified in Table 3.

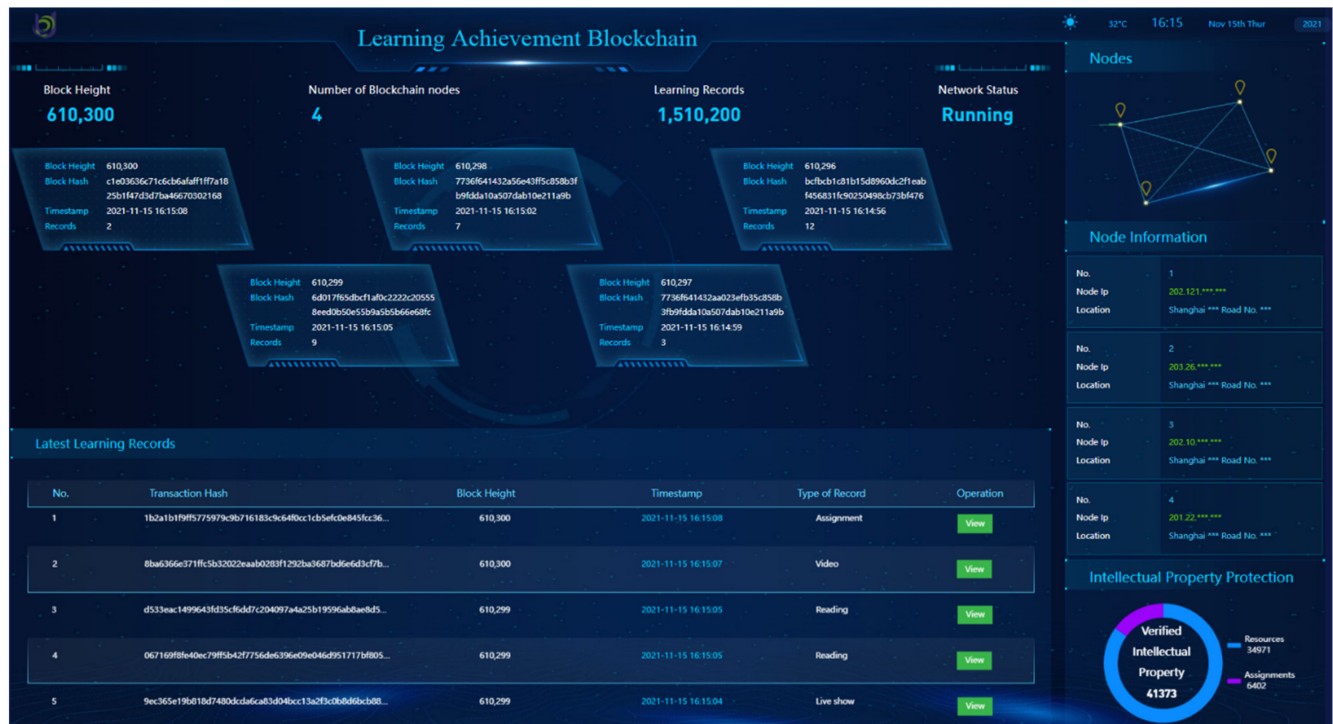

**Figure 7.** Screenshot of proof-of-concept application.

**Table 3.** Key fields of open learning metadata in proof-of-concept.

|  | **UDP** | **UNDP** | **UCM/CCM** |
|---|---|---|---|
| Teacher's profile data | Teacher's platform ID | Teacher's platform ID | Teacher's platform ID |
| Student's profile data | Student's platform ID | Student's platform ID | Student's platform ID |
| Student's learning behavior data | Login timestamp | Login timestamp | / |
|  | Logout timestamp | Logout timestamp | / |
|  | Learning resources ID | Learning resources ID | / |
|  | Learning Span | Learning Span | / |
| Student's learning achievement data | Course submission | Course submission | Course score |
|  | Couse score | Couse score | Credential |

According to the defined open learning metadata, we implemented smart contracts of learning data entry blockchain. Software engineers from the four EdTech vendors implemented data interfaces that extract data from legacy open learning systems and encapsulated these data interfaces into RESTful API services. Our implemented blockchain network invokes these services, parses data, and invokes related smart contracts to accomplish these open learning data entrance of blockchain.

In the front end of the proof-of-concept application, we developed correlation functions that invoke the integrated data and verify their trustability through data-trace function provided by Fabric's smart contract. Based on these correlated data, profiles of students learning in the four legacy HEI systems are created. Recruitment staff of the company can log in the new developed trusted application and ask for interested students' learning data. Students can log in the newly developed trusted application, browse his or her learning data on four legacy open learning systems, and agree to grant access rights to these data to the requesting company or not.

*4.4. Result of the Proof-of-Concept Application*

4.4.1. Functionality

The newly developed "trusted open learning behavior and achievement management" application enables different open learning data originally dispersed in the four separated systems to be collected and stored into the Hyperledger Fabric network, immutably and in a trustworthy manner.

To date, the proof-of-concept system can process not only newly produced learning data, but also the historical learning data, as stated in Section 4.4, i.e., the historical data stored in UDP and UNDP. This illustrates that our proposed blockchain-integrated system bears a strong capacity of historic data traceability in cross-platform open learning and educational resource sharing, which could greatly benefit decentralized HEIs' data interaction by avoiding the hard-achieving pre-defining manner between and among different HEIs.

4.4.2. Performance

We examined the workload of the proof-of-concept system in a production environment over 5 months, and made a statistic on hourly on-chain workloads, which is shown in Figure 8. The peak workload happens at a time between 19:00 and 20:00, and the number of transactions here is 27,286, that is, 7.58 per second on average. In our proof-of-concept application, the submission and feedback timestamp of the on-chain record is written in the off-chain record. It is easy to count the response time of on-chain processing, that is, feedback time minus submission time. We calculated the average time and 95% line time of on-chain response time, which are 1.6 s and 5.6 s, respectively. This performance data illustrates that our proposal is not merely for prototype research; indeed, it is capable for use in industrial production systems.

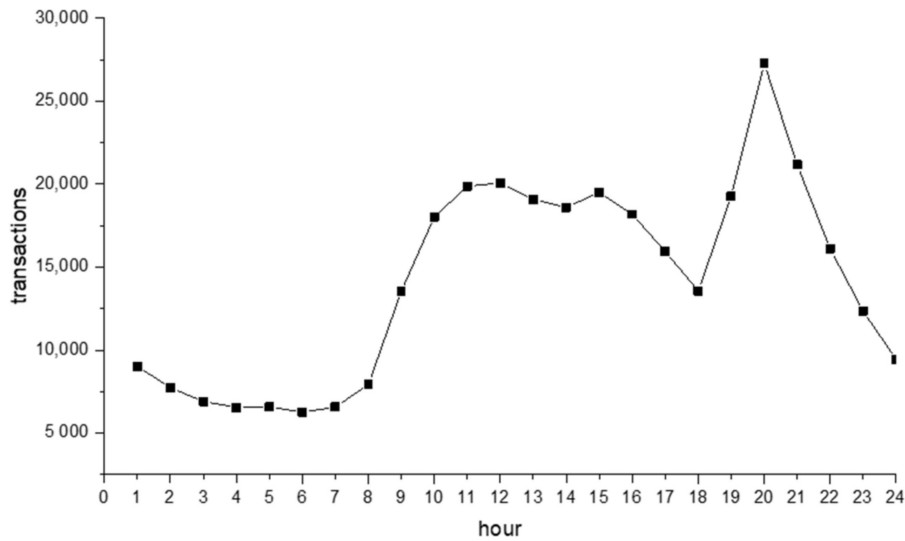

**Figure 8.** Hourly workloads in proof-of-concept application.

**5. Discussion**

In this section, we adopt both qualitative and quantitative methods and make a detailed discussion in terms of our proposed architecture, implementation, and proof-of-concept application. Section 5.1 discusses our system performance of proof-of-concept with extended experiments to illustrate and prove its scalability. Section 5.2 makes a further discussion of our work in comparison with previous works.

*5.1. Scalability Experiment*

In Section 4.4, we presented the performance of the proof-of-concept application which illustrates the competence of our proposal for the production system. However, the

proof-of-concept system is constructed as a 4-node consortium blockchain network and integrates four legacy open learning information systems only. To prove the generality of our proposal, more validation work on scalability needs to be conducted. This validation work can be carried out in two directions: (1) enlarge the quantity of consortium blockchain nodes; and (2) enlarge the quantity of joining legacy open learning systems. Direction 1 falls into the blockchain technology category and involves sophisticated topics such as consensus algorithms, which are beyond the scope of this study. In our research, we assume that with the fixed consortium blockchain implementation version, the increase of blockchain nodes is approximately linear with its performance. This assumption is reasonable because, in an open learning ecosystem, the number of blockchain nodes (which could be corresponded to related legacy open learning systems) usually would not be a large one, as we can define a threshold of 20 (i.e., not exceeding 20) in our research. Direction 2 is the focus of our discussion and needs to be answered with quantitative test data. Technically, more joining legacy open learning systems means a heavier workload of blockchain-entry data. Therefore, we designed a suite of workload testing experiments to analyze whether our framework's scalability could fulfill the pre-defined threshold of 20. In addition, as an illustration for interested readers of direction 1, we also designed a comparison set of running both on 1 blockchain node and on 4 blockchain nodes in our experiments.

These experiments were executed to evaluate the performance of the smart contract invocation (blockchain transaction writes) and query of transaction record (blockchain transaction reads). The testbed system, in line with the production proof-of-concept system, was built upon a blockchain system of Hyperledger fabric 1.4.4 with four organizations. Each organization was served by a virtual machine (node) running orderer. CA and peer services and raft algorithm were adopted among these four orderer services. The hardware configuration of each node in the experiment testbed compared with the production system is shown in Table 4.

**Table 4.** Hardware configuration of each node in experiment testbed and production system.

|  | Experiment Testbed | Production System |
| --- | --- | --- |
| CPU (on each node) | 2 cores | 8 cores |
| RAM (on each node) | 6 G | 32 G |
| Storage (on each node) | 200 G SATA3 | 2 T SSD |

All the four virtual machines in the experiment testbed were allocated in a physical host machine running CentOS 7.5 operation system with one Intel i7-8700K CPU (3.7 GHz 6 cores 12 threads) processor, 32 G RAM and 2 T SATA3 hard disk.

On each node in the experiment testbed, we deployed a unified blockchain restful server to access the underlying fabric blockchain system. To simulate transaction workloads, JMeter was used to generate http requests to Nginx which was running on the host machine and used as a proxy to unified blockchain restful servers.

To test the performance on different workloads, the http requests were generated simultaneously by JMeter with threads quantity of 100, 200, 300, 400, 500 and 600. The Ramp-up period was set to 20 s, and the thread Loop Count was set to 100. We tested both the performance of smart contract invocation (blockchain transaction writes) and query of transaction record (blockchain transaction reads), and measured the system performance by two indexes of throughput and latency. Figure 9 shows the performance of throughput and latency for the smart contract invocation.

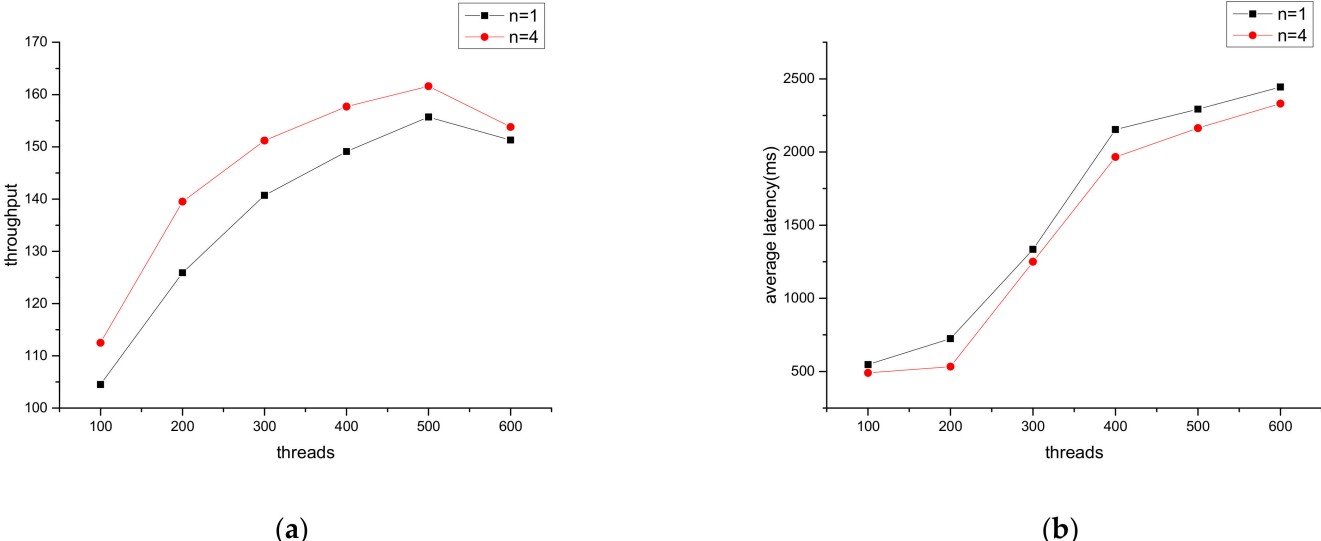

**Figure 9.** Performance of throughput and latency for the smart contract invocation: (**a**) Performance of throughput with one node and four nodes; (**b**) Performance of average latency with one node and four nodes.

In Figure 9, by changing the upstream section of Nginx configuration file, two sets of tests were performed where requests were dispatched to single unified blockchain restful server ($n = 1$) and four unified blockchain restful servers ($n = 4$) in a round robin policy. When the number of threads increased, the throughput grew before it reached the threshold (which is 500 shown in Figure 9), and the average latency grew because the request is more likely to be waiting than processing when more simultaneous requests swarm into the unified blockchain restful server, that is, latency increased due to more time spent waiting.

Like the performance test on the smart contract invocation, Figure 10 shows the results of a performance test of query of transaction record. For the same reason, the average latency grows when the number of threads increases. Because transaction query on node could be processed on local ledger blocks, not relying on global order service, the throughput grew greatly when the number of threads increased from 100 to 200 where $n = 1$, and increased from 100 to 400 where $n = 4$. However, when the number of threads continued to increase, the throughput grew slowly or changed little. While the test was running, we inspected the disk io utility by iostat command on host machine, and found the io utility reached 100% continuously for a long period of time when the number of threads was greater than 400. The results show that after reaching disk io up limits, more simultaneous requests only increase delay without improving the throughput of the query.

By balancing the performance of throughput and latency for the experiments, it was a good choice when threads were 200, where the throughput and average latency of the smart contract invocation were 126 tps and 724 ms, respectively, with $n = 1$. Table 5 shows the comparison of performance on experiment testbed and production system. How the workload and average response time were obtained from the production system is described in Section 4.4. Hardware configuration in experiment testbed and production system is shown in Table 4. The results in Table 5 show that the throughput benchmark in experiment was 17 times more than the workload in the production system while the hardware in the experiment had a much lower configuration than that of the production system. Therefore, the architecture and implementation proposed in this paper can fully satisfy performance requirements.

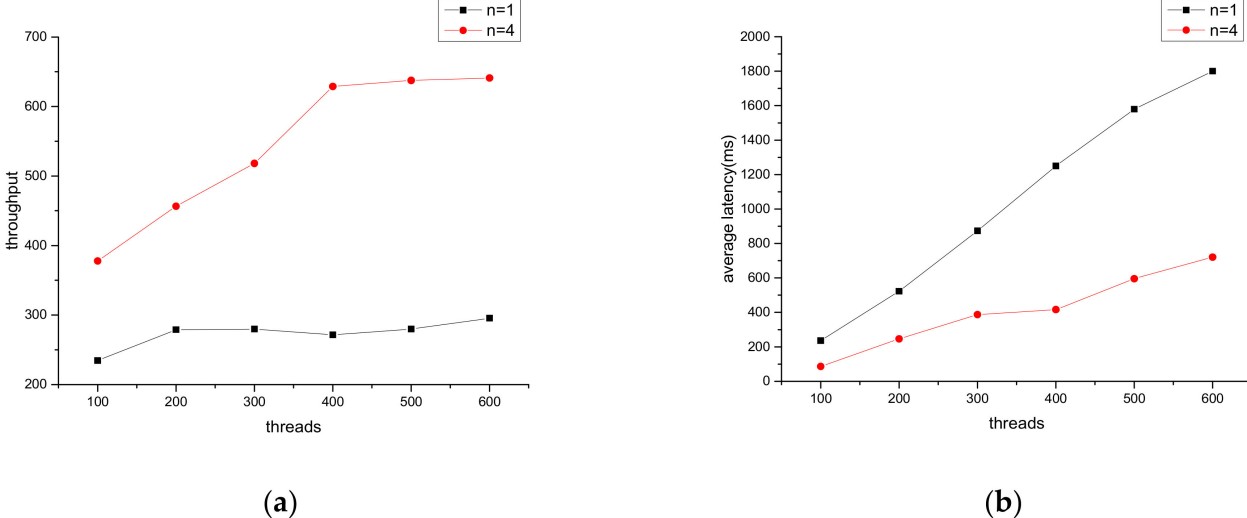

**Figure 10.** Performance of query of transaction record: (**a**) Performance of throughput with one node and four nodes; (**b**) Performance of average latency with one node and four nodes.

**Table 5.** Comparison of performance on experiment testbed and production system.

|  | Experiment Testbed | Production System |
|---|---|---|
| Throughput/Workload | 126 tps (throughput) | 7.58 tps (workload) |
| Average response time | 0.7 s | 1.6 s |

When the system is scaled up, that is, when more legacy open learning systems join our blockchain network, the testbed with lower hardware configuration could still support 17 joined systems, while achieving the same acceptable performance as the proof-of-concept production system. Furthermore, TolFob was implemented upon CrossChain middleware which could dispatch workloads across multiple blockchain networks; thus, by adding more groups of computers and storages, a better performance of throughput can be achieved. Considering these two factors, it is not hard to say that our framework is competent for the assumed threshold of 20 open learning systems joining. Therefore, the architecture and implementation proposed in this paper shows a good extensibility for a data sharing scenario in open learning environments.

### 5.2. Comparison with Previous Work

Section 2 has briefly introduced this study's differentiation from previous works. This section elaborates on a more detailed comparison both qualitatively and quantitatively.

We and previous researchers share a consensus that blockchain technology is a disruptive and promising solution for the open learning data sharing issues. However, due to the complexity of blockchain technology, there are different preferences and views in published papers. Table 6 summarizes these differences.

The first comparison dimension we selected is blockchain type and implementation, i.e., adopting what type of blockchain network in the study. Authors in [4,9,27,28] adopted a consortium blockchain of Hyperledger Fabric implementations (different minor versions due to research time). Authors in [15,16] adopted the public blockchain of Ark and Ethereum. Authors in [2] adopted a proprietary hybrid blockchain. We adopted a consortium blockchain and presented a Hyperledger Fabric-based cross-chain extension.

The second comparison dimension we selected is the integration capacity of the proposed blockchain. All the examined studies except [2] provided smart contract to construct trusted applications. This demonstrates that smart contract has become a fundamental inner schema in blockchain-enhanced application integration. However, only we and

authors in [2,27] provided further out-smart-contract integration function. Moreover, both papers [2,27] only made qualitative discussions of the outer integration function, without presenting a concrete developing framework. We provide a pragmatic software developing framework and implementation that can be directly used as HEI information system integration guideline as well as unified means.

**Table 6.** Comparison of this study with previous works.

| Research Work | Blockchain | Implementation | Smart Contract | Performance | Integration Function |
|---|---|---|---|---|---|
| This study | Consortium | Hyperledger Fabric 1.4 + | Yes | Yes | Yes |
| [2] | Hybrid | N/a | N/a | N/a | Yes |
| [4] | Consortium | Hyperledger Fabric 1.4 | Yes | Yes | No |
| [9] | Consortium | Hyperledger Fabric | Yes | N/a | No |
| [15] | Public | Ark | Yes | N/a | No |
| [16] | Public | Ethereum | Yes | Yes | No |
| [27] | Consortium | Hyperledger Fabric | Yes | N/a | Yes |
| [28] | Consortium | Hyperledger Fabric 1.3 + | Yes | Yes | No |

The paper [2] is a theoretical study on software architecture in an open learning context. This software view is similar to ours. However, authors merely proposed an architecture (combining hybrid blockchain and microservice), made qualitative discussions, and provided no implementation or performance. The paper [4] is a study focusing on blockchain-based secure storage and sharing scheme for MOOCs learning, which is of a narrower but very close context to our study. Similar to us, the authors selected to adopt a consortium blockchain with Hyperledger Fabric 1.4 implementation. However, they primarily focused on smart contract design and deployment, and did not make any cross chain extensions, nor did they provide outer integration function or developing framework, which we did. The case is similar in study [9]. Here, authors focused on the specific theme of Digital Education Resources Authentication. The paper [15] is a study focusing on credit exchange in higher education, which, similarly to encrypted currency, is a very specific theme with a much lower data volume compared to us. This explains the strategy differentiation between ourselves and other authors. They chose to adopt public chains with open-source Ark implementation and did not consider outer integration function. The paper [16] is a study focusing on student's credential sharing. Its research scope and data volume are somewhere between this paper and [15]. Therefore, authors adopted a public chain with Ethereum implementation, and did not consider outer integration function either. The paper [27] is a study focusing on-campus information system integration, which is similar to our integration view. Moreover, these authors adopted a consortium blockchain with Hyperledger Fabric implementation. However, they did not provide a concrete integration framework. The paper [28] is a study of trusted data management, not in the education field but in the context of edge computing. Compared to our open learning context, edge computing involves an IoT network and produces a huge data volume. Authors designed and implemented a Hyperledger Fabric 1.3 extended blockchain, but did not consider the integration issue either.

The third comparison dimension we selected is system or implementation performance. As shown in Table 6, only the three studies [4,16,28] presented their performance data. We carefully examined the data published in these three papers. In study [4], the performance was given in the form of algorithm computational cost, which we cannot compare with our performance of system response time. Meanwhile, studies [16,28] gave their response time of implementation. Comparisons of our study to these papers are listed in Table 7.

**Table 7.** Comparison of response time with related work.

|  | **This Study** | **[28]** | **[16]** |
|---|---|---|---|
| Blockchain platform | Hyperledger Fabric 1.4 + | Hyperledger Fabric 1.3 + | Ethereum |
| Simultaneous requests | 20,000 | 2000 | 1000 |
| Response time(avg)/transaction write | 0.72 s | N/A | 16 s |
| Response time(avg)/transaction query | 0.52 s | 0.79 s | 16 s |

Table 7 shows the response time comparison of our work with the paper [16,28]. Our work is based on Hyperledger Fabric 1.4 and made extensions including cross-chain and integration function. The study [28] was based on Hyperledger Fabric 1.3 and made an extension on security. The study [16] used Ethereum. The data size and experimental configuration were not the exactly same in the three studies. However, as the publishing times are quite close to each other ([16] in 2021 and [28] in 2020), and considering that the configuration we adopted in our study is a common one as of the year 2020, it is reasonable to say that the configuration differentiation would not affect our comparison result too much in this case. To be more persuasive, we selected a higher workload of 20,000 simultaneous requests in our system, compared to 2000 in [28] (10 times) and 1000 in [16] (20 times) to offset the possible configuration bias. The average response time of "transaction query" in [16] was 16 s, which is obviously much larger than 0.79 s (in [28]) and 0.52 s (in our work). The average response time of "transaction write" in [16] was 16 s too, which is also obviously much larger than 0.72 s (in our work, not available in [28]). Authors in [16] also suggested that the reduction of response time could be achieved by using other platforms such as Hyperledger Fabric. These demonstrate that consortium blockchains such as Hyperledger Fabric (which has high computational capabilities and supports time-efficient consensus algorithms) are more suitable for data sharing in an open learning environment than public chains such as Ethereum. The comparison result also indicates that our implementation outperforms other similar works that we have investigated.

To date, previous works have mainly focused on a specific application theme in open learning data sharing with emphasis on features such as security, efficiency, scalability, trusty, etc. Therefore, they leverage blockchain's intrinsic feature and conduct research on smart contacts. However, it is also very important to make the blockchain network open and easier to incorporate various data from different HEI systems. In order to achieve this goal, we adopted both a software architecture and a business feature perspective in this study and implemented Fabric extensions such as CrossChain and out-smart-contract integration functions to make the blockchain a more unified and transparent integration infrastructure to legacy HEI systems, while other works had no such feature and found it difficult to handle this complicated open learning system interoperation scenario.

## 6. Conclusions and Future Work

To resolve the cumbersome interoperability issue of authentic data sharing among the open learning education ecosystem, a consortium blockchain is leveraged and extended in our study. The most vital part of our research is to propose an overall architecture consisting of an open learning business schema, a conceptual application model, and a pragmatic blockchain integration framework, as a guideline and infrastructure for a blockchain-enhanced trusted open learning application development. The results of our implementation and proof-of-concept indicate that our consortium blockchain extended framework is competent for multiple HEI open learning systems integration, with an average response time of 1.6 s when no more than 20 systems were integrated. To the best of our knowledge, this result outperforms other research findings we have investigated. Based on these, it can be said that, under the assumption that the quantity of related information systems in a specific open learning ecosystem usually would not surpass 20, the proposed architecture and framework bear the potential to be widely adopted in open

learning data sharing scenarios, as a trusted and unified interoperation infrastructure to create better open learning platforms and bridge the gap in the present open learning ecosystem. This further implies that our research finding may play an important role in the forthcoming significant opportunity in the creation of disruptive open learning business models and flexible open learning ecosystems with the disruptive features of blockchain technology.

This research work still has some limitations. Blockchain in education is essentially for stakeholders of the ecosystem to establish standardization and validation of HEI educational systems to mitigate fraud. In this regard, the limitation of our work mainly lies in two aspects: a lack of data governance specification and cross-chain standards. In order to provide a unified open learning data-sharing infrastructure solution, there should be a specification of data governance for all stakeholders to abide by when integrating legacy open learning systems, especially for the procedure or workflow consensus. Furthermore, a cross-chain standard would be more efficient and elegant for the interoperation of blockchain. Our work focuses on application architecture and software framework, while simplifying these two aspects by proposing weak substitute versions.

In light of this study's limitations, in the future, we will conduct further research work, which includes proposing a more complete crosschain protocol and core component implementation, adapting more open-source consortium blockchain implementations (besides Hyperledger Fabric), inviting more-open learning HEI stakeholders connecting to our blockchain network, and promoting a team standard for open learning system data governance.

**Author Contributions:** Conceptualization, J.X. and Y.J.; methodology, J.X.; software, Y.L.; validation, J.X., Y.J. and Z.J.; resources, J.X.; data curation, Y.L.; writing—original draft preparation, Y.J., J.X., Y.L. and Z.J.; writing—review and editing, Y.J., J.X., Y.L. and Z.J.; project administration, J.X. and Y.J.; funding acquisition, J.X. All authors have read and agreed to the published version of the manuscript.

**Funding:** This study was supported both by Shanghai Science and Technology Innovation Action Plan International Cooperation project "Research on international multi language online learning platform and key technologies (No.20510780100)" and Science and Technology Commission of Shanghai Municipality research project "Shanghai Engineering Research Centre of Open Distance Education (No.13DZ2252200)".

**Institutional Review Board Statement:** Not applicable.

**Informed Consent Statement:** Not applicable.

**Data Availability Statement:** The data presented in this study are available on request from the corresponding author. The data are not publicly available due to their usage in a production system.

**Conflicts of Interest:** No potential conflict of interest was reported by the authors.

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
