# Peer review of "Towards a Trusted and Unified Consortium-Blockchain-Based Data Sharing Infrastructure for Open Learning—TolFob Architecture and Implementation"

_sustainability, doi:10.3390/su132414069_

Round 1
Reviewer 1 Report
This is a very important topic and worth researching, nonetheless in order to make the paper publishable I would ask you to include the following changes:
1) In the article you mention "open learning", but I think you should introduce also the term "open learning resources".
2) When you discuss the software platform I think that you should mention the most popular MOOC ecosystem (eg. edX, Coursera, Udatity, FutueLearn), which is so popular nowadays, see for example: https://ier.uek.krakow.pl/index.php/pm/article/view/1763
3) At the end of point 5 (or in point 6) I would welcome the scientific discussion that is the comparison of your results with prior researchers and to position your contribution. Are your results in line with previous articles? Is it against? Is it sth new?
4) When discussing and comparing your results with previous other authors you may refer to: https://doi.org/10.4467/2353737XCT.19.026.10162
5) In the Conclusions section (point 6) of your article please remember that we need to have four compulsory elements:
(i) general summary of empirical results,
(ii) implications for practice,
(iii) research Limitations
and (iv) suggestions for further research in this topic.
I really miss some of them (e.g. limitations). Please complete it.
6) The submission lacks research questions (RQ), which are essential in all submissions. Maybe it will be good to ask the RQ in your introduction?
7) Please proofread the article, it is not free of mistakes and typos.
I hope these comments will be useful for you. Good luck with revising your paper.
Author Response
These comments are indeed very helpful to us. We the authors are very thankful for your precious time and revision suggestions. Please see the attachment.

Reviewer 2 Report
Thank you for a comprehensive, exciting paper. My recommendations:
- It is not possible to construct this kind of graph based on 4 points. On what assumption did the authors decide that this would be a connected graph? It is evident that the graphs (Figure 10 - 21) do not capture linear dependencies.
- The graphical design of Figures (1-6 and 9) is graphically weak, and I recommend that they be redesigned. Figure 8 is inconsistent with the other graphs. I think all figures should be displayed in one style.
- The Paper is overall not clear enough, does not work with the IMRAD structure.
- It offers technical details, but it is not clear what the implications are concerning education. Therefore, I highly recommend including an educationally focused discussion section. The current discussion just summarises the previous one, with no relation to other literature or educational issues.
- The authors answer four questions, isn't that a lot? Wouldn't it be better to split the text? I leave this for consideration, but it would make the text much more transparent for me as a reader.
- I appreciate the very well done introduction and the excellent work with relevant literature.
- I appreciate the capture of the whole implementation process (from design to testing) of the technical solution.
- I would have appreciated a more significant relation of the text to education, which the authors undeniably have and understand, but they do not put enough of it in the text.
- Some work needs to be done on formatting, especially on inserting figures and indenting them.
- Some references DOI are blue and underlined, others formatted to be just black with no underlining. This needs to be standardized.
- There are typographical errors in the figures (e.g. missing spaces) that need to be corrected. Overall, the comments on the graphs in the text are descriptive. It is questionable whether this part of the paper could be reduced.
Author Response

(The authors gave the same response as above.)

Round 2
Reviewer 2 Report
An old version of the abstract remains in the system, which does not correspond to the new version from the PDF (it also contains a typographical error). It must be modified before publication. I appreciate the changes made and their incorporation. Thank you. Good job.
Author Response
We appreciate the reviewer’s suggestion very much. We had made some minor revisions, specifically in line 427, line 608, and line 712. We would like to know whether these refer to the typographical errors mentioned by the reviewer. If not, we would be grateful if the reviewer could tell us the errors.
